# The Vaginal Virome—Balancing Female Genital Tract Bacteriome, Mucosal Immunity, and Sexual and Reproductive Health Outcomes?

**DOI:** 10.3390/v12080832

**Published:** 2020-07-30

**Authors:** Anna-Ursula Happel, Arvind Varsani, Christina Balle, Jo-Ann Passmore, Heather Jaspan

**Affiliations:** 1Department of Pathology, Institute of Infectious Diseases and Molecular Medicine, University of Cape Town, Anzio Road, Observatory, Cape Town 7925, South Africa; christina.balle@uct.ac.za (C.B.); jo-ann.passmore@uct.ac.za (J.-A.P.); hbjaspan@gmail.com (H.J.); 2The Biodesign Center of Fundamental and Applied Microbiomics, School of Life Sciences, Center for Evolution and Medicine, Arizona State University, 1001 S. McAllister Ave, Tempe, AZ 85287-5001, USA; Arvind.Varsani@asu.edu; 3Structural Biology Research Unit, Department of Integrative Biomedical Sciences, Institute of Infectious Diseases and Molecular Medicine, University of Cape Town, Anzio Road, Observatory, Cape Town 7925, South Africa; 4NRF-DST CAPRISA Centre of Excellence in HIV Prevention, 719 Umbilo Road, Congella, Durban 4013, South Africa; 5National Health Laboratory Service, Anzio Road, Observatory, Cape Town 7925, South Africa; 6Department of Pediatrics and Global Health, University of Washington, 1510 San Juan Road NE, Seattle, WA 98195, USA; 7Seattle Children’s Research Institute, 307 Westlake Ave N, Seattle, WA 98109, USA

**Keywords:** vaginal virome, female genital tract, microbiota, bacteriome, host immunity, adverse birth outcomes

## Abstract

Besides bacteria, fungi, protists and archaea, the vaginal ecosystem also contains a range of prokaryote- and eukaryote-infecting viruses, which are collectively referred to as the “virome”. Despite its well-described role in the gut and other environmental niches, the vaginal virome remains understudied. With a focus on sexual and reproductive health, we summarize the currently known components of the vaginal virome, its relationship with other constituents of the vaginal microbiota and its association with adverse health outcomes. While a range of eukaryote-infecting viruses has been described to be present in the female genital tract (FGT), few prokaryote-infecting viruses have been described. Literature suggests that various vaginal viruses interact with vaginal bacterial microbiota and host immunity and that any imbalance thereof may contribute to the risk of adverse reproductive health outcomes, including infertility and adverse birth outcomes. Current limitations of vaginal virome research include experimental and analytical constraints. Considering the vaginal virome may represent the missing link in our understanding of the relationship between FGT bacteria, mucosal immunity, and adverse sexual and reproductive health outcomes, future studies evaluating the vaginal microbiome and its population dynamics holistically will be important for understanding the role of the vaginal virome in balancing health and disease.

## 1. Introduction

Despite the considerable literature published on bacterial communities, only a few human microbiota studies have focused on viruses, fungi, protists and archaea communities [1,2,3], yet shifts in one community likely modulate community structure of others and vice versa. It is estimated that only about 1% of the human virome has been described at the sequence level [4], and even less has been characterized functionally and hardly any studies have evaluated the virome in relation to reproductive health and the female genital tract (FGT).

From various biological niches, such as the human gut, we know that viruses interact with other components of the microbiota and the human host, consequently influencing human health [5,6,7,8]. However, little is known about the interactions of prokaryote- and eukaryote-infecting DNA and RNA viruses present in the FGT microbiota, collectively making up the vaginal virome, with other components of the vaginal microbiota or the human host, and its consequent impact on health outcomes. The primary outcome of this review was to summarize our current understanding of the pro- and eukaryote-infecting viruses making up the vaginal virome. Secondary outcomes included describing interactions between the vaginal virome and other constituents of the vaginal microbiota, and possible associated adverse health outcomes. The literature search for this non-systematic review was conducted on 31 March 2020, using two databases (PubMed and GoogleScholar), with the search terms (“vagina” OR “female genital tract” OR “female reproductive tract”) AND (“virome” OR “virus” OR “viral” OR “microbiome”) and was limited to studies that were in published in English.

## 2. Known Components of the Vaginal Virome

### 2.1. Eukaryote-Infecting Viruses

A range of vaginal DNA viruses infecting eukaryote cells have been identified by shotgun metagenomics of vaginal samples from generally healthy, asymptomatic women of reproductive age participating in the Human Microbiome Project (HMP) [9], including double-stranded (ds) DNA (families *Adenoviridae*, *Herpesviridae*, *Papillomaviridae* and *Polyomaviridae*) and single-stranded (ss) DNA viruses (families *Anelloviridae*) (Table 1 and Figure 1). The most common viruses detected in the lower reproductive tract were alphapapillomaviruses, with 38% of the participants being infected with at least one alphapapillomavirus [9]. To date, more than 220 human papillomavirus (HPV) types have been identified, including at least 50 that preferentially infect the genital mucosa [10,11] (Table 1 and Figure 1). Longitudinal sampling in the HMP suggests that up to 50% of these papillomaviruses establish productive infections and were replication competent, as many of the viruses were detected over multiple time points in the same women. Additional dsDNA viruses have been identified in vaginal samples from pregnant women and women with reproductive disorders [12,13,14] that share sequence similarities to those in the families *Alloherpesviridae, Iridoviridae, Marseilleviridae, Mimiviridae, Phycodnaviridae* and *Poxviridae* (Table 1 and Figure 1). In a cohort of women undergoing in vitro fertilisation, *Herpesviridae*, *Polyomaviridae*, *Papillomaviridae* and *Anelloviridae* were present, with *Papillomaviridae* being the most common virus family detected [15], while in a cohort of sixty pregnant women, anelloviruses were the most commonly detected viruses in vaginal samples, being present in 42% of women screened [12]. In a cohort of women coinfected with human immunodeficiency virus (HIV) and HPV, four viral families were identified in the FGT: *Papillomaviridae, Anelloviridae, Genomoviridae* and *Herpesviridae* [16]. Papillomavirus reads were more abundant in women with premalignant cervical lesions, which were also strongly associated with carrying multiple high-risk HPVs, while anellovirus read abundance was negatively correlated with host CD4+ T-cell counts [16]. Similarly, 46 known HPV types were detected in the FGT of women living with HIV, in addition to viruses belonging to the *Polyoma-* and *Anelloviridae* families [17]. Whether a core vaginal virome exists or whether differences in vaginal viruses identified in these studies are due to variable demographic or clinical characteristics of the cohorts, to differences in laboratory or sequencing methods, or to the viral databases and viral annotation tools applied remains unclear. To our knowledge, only one sequencing study [13] reported RNA viruses in the reproductive tract, belonging to the *Partitiviridae* family (dsRNA), of which fungi is the natural host. This is likely to be due to the poor stability of stored RNA and more complex laboratory assays being required to sequence RNA viruses. HIV-1, an ssRNA reverse transcribing virus (ssRNA-RT) in the family of *Retroviridae*, is also detectable in FGT secretions of women living with HIV during viral shedding [18,19,20].

While none of the women participating in the HMP had any genital symptoms, the high prevalence of eukaryote-infecting vaginal viruses raises the question of whether these vaginal viruses play a role in reproductive health. As the majority of humans remain asymptomatic to some viral infections, it has been proposed that viruses have become part of the metagenome of “healthy” individuals, rarely causing disease and remaining dormant within the host [21]. However, some well-described disease-causing viruses, such as high-risk HPV types, herpes simplex virus (HSV)-2 and polyomaviruses were also described to be part of the vaginal virome of some individuals despite being asymptomatic (Table 1 and Figure 1). Furthermore, viruses that do not overtly cause disease yet establish chronic infections have been shown to influence immunity at other mucosal surfaces, such as the gut [22,23], and this is also likely to occur in the lower reproductive tract and is thus discussed in more detail below.

### 2.2. Prokaryote-Infecting Viruses

Few shotgun metagenomic studies have investigated the presence or function of prokaryotic viruses in the lower reproductive tract of women. Although prokaryotic-infecting viruses, from now on referred to as bacteriophages, are estimated to be amongst the most abundant living entities on Earth [24] and are thought to play an important role in shaping the bacterial microbiota and associated health outcomes in the human gut [5,25,26,27], oral cavity [28,29], skin [30,31] and lungs [32], their role in the lower reproductive tract is understudied. Functionally, bacteriophages are divided into lytic (virulent) and temperate types based on their differential ability to either lyse host cells (to release progeny bacteriophages) or to incorporate their genomes into the host cell genome as prophages and remain dormant, respectively [33]. Several groups have identified functional and nonfunctional prophages in the genomes of vaginal bacterial species [34,35,36,37,38]. Strong bioinformatic and in vitro evidence indicates that vaginal *Lactobacillus* strains (including *L. crispatus, L. gasseri, L. jensenii* and *L. plantarum* isolates) carry inducible prophages [34,35,36,38]. One third of all vaginal *Lactobacillus* strains from South African women that have been examined to date were found to harbour at least one prophage, with *L. crispatus* more commonly harbouring prophages than *L. jensenii* [36]. For most of these *Lactobacillus* prophages, however, factors determining their induction and permissive bacterial hosts are yet to be determined.

Prophages have also been described in the genomes of vaginal and urinary bacterial species that have been associated with adverse reproductive health outcomes, including *Gardnerella vaginalis* [37], Group B *Streptococcus* (GBS) [39] and *Enterococcus* spp. [40]. Among 39 *Gardnerella* strains, a species associated with bacterial vaginosis (BV), more than 400 annotated prophage sequences have been identified and evidence of ongoing prophage acquisition within these *Gardnerella* populations was present [37]. Another study found that almost 90% of the examined genomes of vaginal and urinary *Gardnerella* strains contained at least one prophage sequence [40]. Similarly, almost 80% of GBS isolates had at least one prophage, which carried genes encoding factors previously associated with host adaptation and virulence [39]. The high abundance of prophage sequences within the genomes of vaginal bacterial species suggests that bacteriophages might play a role in shaping the bacterial microbiota of the FGT and associated health outcomes, similar to other biological niches. Recent metagenomic sequencing of vaginal samples revealed that the majority of identified vaginal DNA viruses are dsDNA bacteriophages, similar to those in the families *Myo-, Podo- and Siphoviridae* [14]. In addition, various unclassified viruses within the order *Caudovirales* were found to be present. Notably, only 4% of vaginal viruses identified by metagenomic sequencing by Jakobsen et al. (2019) targeted eukaryotes [14], confirming the high abundance of prokaryote-infecting viruses within the FGT microbiota (Table 1 and Figure 1). Considering that a shift from lysogeny to a lytic lifecycle has been correlated with disease within the gut microbiota [41], the role of bacteriophages for FGT health remains a crucially important yet underexplored area.

## 3. Interaction of the Vaginal Virome with Other Components of the Vaginal Microbiota and Human Host

### 3.1. Interactions between the Viral and Bacterial Microbiota

Limited studies have examined the interactions between all components of the vaginal microbiota and the human host. Observational studies have shown that the acquisition and transmission of viral sexually transmitted infections (STIs), including HSV-2, HPV and HIV, are more common in women with high diversity, nonoptimal vaginal bacterial microbiota [42,43,44]. However, many studies did not adjust for confounders to this relationship, such as the presence of other STIs, sexual risk behaviour (condomless sex) and circumcision status of sexual partners. Recent evidence from longitudinal studies suggests that changes in the bacterial microbiota and associated immunomodulatory metabolites precede incidental STIs [45] and have been associated with a higher rate of HPV persistence and genital HSV-2 shedding [43]. Data from the HMP further indicated that alpha-papillomaviruses were more common in women with a high vaginal bacterial diversity than in those who had a *Lactobacillus*-dominant microbiota [9]. Strong co-abundances between bacteriophages and predicted bacterial hosts were observed in a metagenomic study [14], further suggesting that viral and bacterial communities interact within the lower reproductive tract. As such, links between viral community composition and the presence of *L. crispatus*, *L. iners, G, vaginalis* and *A. vaginae* in the FGT were found [14]. Diversity changes in the vaginal eukaryotic DNA virome over the course of pregnancy appeared similar to concomitant changes in the vaginal bacteriome, and pregnant women with highly diverse vaginal viromes tended to also have highly diverse vaginal bacteriomes [12]. While a relationship between bacterial and viral communities within the human lower reproductive tract appears to be supported by these studies, the directionality of this relationship is not clear and causality is yet to be established. It remains to be determined if changes in the bacterial microbiota are driven by changes in the viral community structure, or if susceptibility, persistence and clearance of viral infections is influenced by the vaginal bacterial microbiota. It is also important to acknowledge that underlying indirect mechanisms may regulate both, bacterial and viral communities, in addition to a more direct microbial mechanism implied by available studies.

### 3.2. Interactions between the Viral and Fungal Microbiota

There is clinical and in vitro evidence that the vaginal bacterial microbiota influences colonisation with fungal strains [46,47], albeit studies investigating yeast–viral interactions in the FGT are limited. The majority of available research of yeast–viral interactions focuses on the relationship between viral STIs and *Candida* coinfections, in which vaginal *Candida* infection was investigated as a risk factor for HIV transmission [48,49]. However, this is likely to be the results of mucosal barrier disruption and/or inflammation associated with vulvovaginal candidiasis. Seventy-five percent of women experience at least one episode of candidiasis during their life, and the risk for frequent, more invasive and resistant infections in persons living with HIV is high, likely due to T cell immune defects [50]. Other in vitro studies have shown that HSV-2 significantly enhances binding of *Candida albicans* to HeLa cells [51], which the authors conclude suggests that HSV-2 increases *Candida* persistence in the FGT. Interestingly, in vitro studies have also shown that HSV-1 and coxsackievirus-B5 were retained in and released by *C. albicans* biofilms, in which the viruses remained viable and protected from antiviral agents as well host factors [52]. *Pseudomonas* phages have been described to inhibit *C. albicans* biofilms as well as their planktonic growth [53]. Whether yeast-infecting bacteriophages are commonly present in the vagina is unknown, although as mentioned previously, *Partitiviridae*, typically a fungal phage, have been identified in the female reproductive tract [13]. In turn, HSV-1 has been shown to protect *C. albicans* by downregulating monocyte-mediated anti-*Candida* immune system responses [54], suggesting bilateral interactions between fungal and viral communities in the FGT. HIV-1 envelope and transactivating proteins have also been shown to bind to *C. albicans*, which may promote fungal virulence by inducing hyphae formation [55,56,57]. Although the clinical relevance of these in vitro results remains to be elucidated and research on a broader range of vaginal viruses and fungi is needed, these studies suggest that there may be clinically relevant interactions between viral and fungal communities in the FGT and that an imbalance between these may influence reproductive health.

### 3.3. Interactions between Virome and Host Immunity

Microbial communities also interact with the host and influence innate immunity, immune cell activation and cytokine secretion. A balanced interplay between the vaginal microbiome and host immunity is crucial to prevent infections on the one hand but to maintain an immunotolerant environment, particularly during pregnancy, on the other hand. While the interaction of the vaginal bacteriome with the host has been reviewed by others [43,58,59], there are less data available on the effect of eukaryote-infecting viruses other than viral STIs, prokaryote-infecting viruses and the collective virome on host immunity. The innate immune system, including epithelial cells and mucus, Toll-like receptors, antimicrobial peptides and defensins, cytokines and innate immune cells, is an important host immune defence mechanism in the FGT, collectively minimizing the risk of viral infection upon exposure [60,61]. Similarly, cellular and humoral adaptive immune responses to viral STIs have been described in detail. While HIV-1 infection leads to induction of CD8+ T-cell responses in the cervical mucosa [61], HSV-2 infection has been associated with cervical CD4+ T cell numbers and a distinct cytokine profile including various cytokines without significant alterations in local proinflammatory cytokines [42], and HPV infection has been associated with a T helper (Th) 1 cytokine immune response and elevated proinflammatory and regulatory cytokines [62,63,64]. In germ-free mice, expansion of enteric bacteriophages has been linked to immune cell expansion and increased inflammation in the gut, and phage DNA isolated from human faeces stimulated interferon (IFN)-γ production by dendritic cells in mice [8]. Further, certain viruses may be important for conditioning of immune responses. In the gut, infection by a persistent strain of murine norovirus compensated for the absence of bacteria in germ-free mice by restoring intestinal morphology and by promoting lymphocyte differentiation [22]. The same virus protected mice from lung injury following infection with *Pseudomonas aeruginosa* and restored serum immunoglobulins in germ-free mice to levels observed in conventional mice [65], yet the relevance of these findings for humans and specifically the FGT remains to be confirmed.

These initial studies point towards an important role of the vaginal virome in innate and adaptive immunity of the FGT. While many of these initial findings are based primarily on disease-associated viruses, they may suggest that changes in the collective virome may similarly lead to altered mucosal immunity in the FGT and thus may impact sexual and reproductive health outcomes.

## 4. The Vaginal Virome and Adverse Sexual and Reproductive Outcomes

### 4.1. Eukaryotic-Infecting Viruses

As reviewed elsewhere, sexually transmitted viruses, like HSV-2, HPV, cytomegalovirus (CMV), hepatitis B or HIV, have been associated with a range of adverse health outcomes [66,67,68,69,70,71], including cervical cancer (HPV) [66], genital ulcers (HSV-2), aseptic meningitis as well as vertical infections in infants, such as neonatal herpes (HSV-2) [67] or cirrhosis and liver cancer (hepatitis B) [69,70].

### 4.2. Bacterial Vaginosis

BV is a risk factor for severe reproductive complications [72,73,74] and STIs [49,75,76], and it has been suggested by several authors [34,77,78] that bacteriophages are a contributing underlying biological cause for the rapid change in composition of vaginal bacterial communities associated with onset of BV and its difficulty to treat. With advances in shotgun metagenomics, further evidence is emerging that eukaryote- and prokaryote-infecting DNA viruses, including bacteriophage communities, may differ between women with and without BV [14], although others previously did not find any differences by vaginal bacterial community state type [44], possibly due to different laboratory and analysis methods used. Jakobsen et al. (2019) found that, while no significant difference in overall viral alpha diversity was present between groups, bacteriophage operational taxonomic units (OTUs) and predicted bacterial host OTUs strongly correlated in women with BV. Miller-Ensminger et al. (2018) observed variations between the prophage populations of women with and without overreactive bladder symptoms, suggesting that bacteriophages may contribute to genitourinary health [40].

### 4.3. Infertility

Infertility is defined by the failure to establish pregnancy after 12 months of regular sexual intercourse. While the most predictive factor for female infertility is a woman’s age, biological and environmental factors are believed to contribute as well [79]. Changes in the vaginal bacterial microbiota have been described as a risk factor for infertility [80,81,82], and it is likely that viruses, such as human herpesviruses (HHV), also play a role. In up to 20% of infertile couples, the male or female partner had urogenital bacterial infections [83], and in agreement with older women being more likely to be infertile, it has been described that seroprevalence of HHV-8, which similarly to HHV-6 can be transmitted via saliva, is increasing with increasing age [84]. HHV-6 DNA was found in 43% of endometrial epithelial cells from infertile women in contrast to being completely absent from the endometrium of fertile women [85,86]. HHV-6A-specific endometrial Natural Killer (NK) cell numbers and cytokine responses were elevated in women who had HHV-6A present [85], suggesting that HHV-6 infections may modify endometrial immune cell and inflammatory profiles, resulting in the inability to sustain a successful pregnancy. Further, endometrial NK cells had increased expression of several chemokine receptors and endometrial epithelial cells upmodulated the corresponding ligands, including Monocyte Chemotactic Protein 1 (MCP1 and CCL2), Interferon Gamma-induced Protein 10 (IP-10 and CXCL10) and Eotaxin-3 (CCL26) [86]. Others assessed whether HHV-6 played a role in preventing embryo implantation and found that 50% of women with two or more failed embryo transfers had detectable HHV-6 late viral proteins in endometrial epithelial cells and that women who were HHV-6 positive had undergone significantly more failed transfers than those who were HHV-6 negative [87], further providing evidence that HHV-6 infection might influence fertility and pregnancy outcomes.

In subfertile women undergoing in vitro fertilisation with a fresh embryo transfer, an association of the eukaryotic vaginal virome with prophylactic antibiotic exposure and reproductive outcomes has been described [15]. While there was no association between viral diversity and clinical pregnancy overall, a higher diversity of herpesviruses and alphapapillomaviruses was present in women who received prophylactic azithromycin treatment compared to women not receiving azithromycin. Further, in women receiving azithromycin, viral diversity was higher in women whose embryo transfer did not result in clinical pregnancy compared with those who achieved clinical pregnancy [15], suggesting that both bacterial and viral components of the vaginal microbiota may influence the ability to achieve clinical pregnancy.

### 4.4. Adverse Birth Outcomes

A number of clinically relevant viruses can be vertically transmitted from mothers to their foetus, including Zika virus, HIV, hepatitis B virus, hepatitis C virus, HSV-1 and -2, varicella zoster virus, Rubella virus, parvovirus B19 and CMV, and can cause stillbirth or severe morbidity in infants [88,89,90,91,92,93,94]. In the context of preterm birth (PTB) and small-for-gestational age (SGA) births, biological mechanisms are still poorly understood. Sequence-based studies of the vaginal bacterial microbiota have revealed that high-diversity bacterial communities [95,96,97] and possibly increased concentrations of vaginal inflammatory markers [95,98] might contribute to increased risk for these adverse birth outcomes (ABOs). Recently, a higher richness of the vaginal eukaryotic DNA virome, including viral sequences similar to those in the families *Adenoviridae, Anelloviridae, Herpesviridae, Papillomaviridae, Polyomaviridae* and *Poxviridae,* has been linked to spontaneous and medically indicated PTB in North American women, but not one specific virus or group of viruses could be linked to PTB [12]. Having both high bacterial and viral diversity in the first trimester of pregnancy was linked to the highest risk for PTB, indicating that the interplay of bacterial and viral communities or an imbalance thereof may be a mechanism by which PTB is triggered. Neither whether higher viral diversity would also be associated with PTB in other populations nor how demographic factors influence the vaginal virome during pregnancy, are clear.

Focusing on specific viral groups rather than the collective virome, vaginal HPV infections have been linked to SGA and low birth weight in observational studies, independently of other risk factors [99,100]. In silico analyses have revealed significant associations between invasive neonatal-infecting GBS isolates and harbouring of a specific group of prophages within their genomes [39]. Various viruses were also identified in amniotic fluid samples obtained from pregnancies with adverse outcomes, with adenovirus, CMV and enterovirus being the most common, while few healthy pregnancy controls had any virus detected [101]. More than half of women delivering infants with intrauterine growth restriction had viruses in their amniotic fluid during pregnancy, and adenovirus was detected in the amniotic fluid collected by amniocentesis in 60% of these women [101]. Culture- and sequencing-based studies have found evidence of bacterial colonisation in amniotic fluid from pregnancies with ABOs [102,103,104], suggesting that similarly viruses might be present in amniotic fluid in pregnancies with adverse outcomes, where anatomical, physiological or immunological barriers are compromised. It remains to be determined whether bacterial and viral communities are present in amniotic fluid from healthy pregnancies, as recent sequencing-based studies have drawn conflicting conclusions [105,106,107,108].

It also has been shown in vitro that adenoviral infection of extravillous trophoblast cells in the presence of maternal decidual lymphocytes induces trophoblast apoptosis [109], indicating that the maternal inflammatory response to adenovirus might induce placental cell death and subsequent ABO. In mouse models of pregnancy, however, a combination of bacteria and eukaryotic-infecting viruses induced PTB, but neither alone was capable of causing PTB [110,111]. Similarly, viral infection of the murine cervix and placenta has been shown to alter the inflammatory responses to subsequent bacterial infection of the FGT [110,111,112], indicating that viral infection of the FGT during pregnancy may alter the capacity to control ascending bacterial infections, which subsequently may lead to ABO.

Data relating to severe acute respiratory syndrome coronavirus 2 (SARS-CoV-2) infection of the FGT and the consequent relationship with pregnancy outcomes is sparse but continually accruing. A recent systematic review and meta-analysis of 79 pregnant women with coronavirus-related symptoms, of which half had confirmed SARS-CoV-2 infection, reported a significant prevalence of placenta-mediated disorders with high rates of miscarriage, PTB, preeclampsia and foetal growth restrictions [113]. While there is currently still conflicting yet continuously accumulating evidence for the presence of SARS-CoV-2 in vaginal secretions, amniotic fluid, cord blood or breastmilk in infected women [114,115,116,117], infection and visualization of SARS-CoV-2 in placental tissue has been demonstrated thoroughly [116,118]. The first case of transplacental transmission of SARS-CoV-2 from a pregnant woman affected by COVID-19 during late pregnancy to her neonate has been described [116], suggesting potential perinatal SARS-CoV-2 transmission. Further investigations are warranted to confirm vertical transmission of SARS-CoV-2 and long-term consequences thereof.

## 5. Limitations of Current Research

The growing interest in the field of vaginal virome requires standardisation of laboratory protocols and analysis pipelines, including identification of RNA viruses, adequate use of negative controls to account for contamination with environmental viruses, continued development of high-throughput sequencing accessibility, and advancement in viral annotation databases and tools. Adequate use of viral nucleic acid extraction methods or kits that have not been associated with contamination as well as removal of bacterial and human nucleic acids either during the extraction process or after sequencing are crucial. It is concerning that various sequencing-based studies have described the presence of viruses in the FGT that share similarities to nonhuman, nonbacterial and nonfungal virus families. For example, vaginal *Iridoviridae* have been described, which were previously only thought to infect ectothermic vertebrates, insects and crustaceans [119]. Similarly, *Phycodnaviridae* were also described to be present in vaginal samples, which were previously only thought to infect algae [120]. While this suggests that these viral sequences might have been present due to contamination or might have been incorrectly taxonomically classified or that they share similarities to sequences of viruses in these families (since there are clearly protein homologues found across viral families, especially the RNA-dependent RNA polymerases, helicase domains of replication proteins and, in some cases, capsid proteins [121]), it might be possible that these findings are real due to vaginal insertional or hygiene practices. Further, analysis of low biomass samples, such as breastmilk, placenta or amniotic fluid, requires rigorous use of controls, including nucleic acid extraction and PCR controls, as well as environmental swabs collected and processed as samples as additional negative controls.

## 6. Conclusions and Future Perspectives

There is a paucity of research on the vaginal virome and the interplay of vaginal bacterial and viral communities and host immunity and its likely effect on sexual and reproductive health outcomes. Once systematic laboratory and analysis pipelines have been established for both DNA and particularly RNA viruses, the vaginal virome should be characterized in broader populations, such as pre- and postmenopausal women and women from different demographics, and in connection with other components of the vaginal microbiota and the human host. Functional characterization of viruses present in the vaginal virome and evaluation of their effect on reproductive and sexual health outcomes are crucial, and highly detailed longitudinal clinical cohorts with frequent sampling or animal models will be required to assess causality.

## Figures and Tables

**Figure 1 viruses-12-00832-f001:**
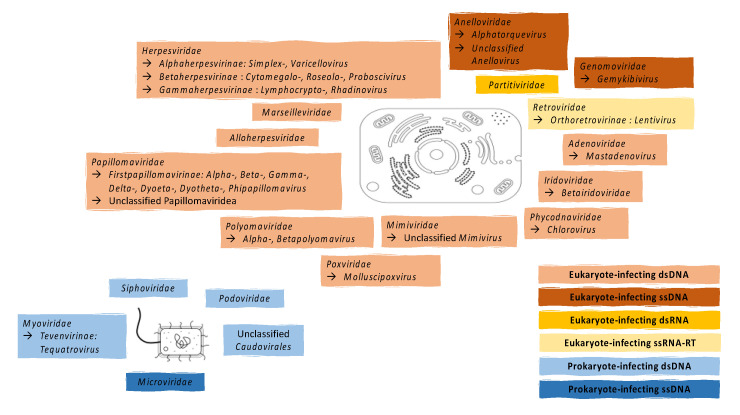
Current knowledge on eukaryote- and prokaryote-infecting vaginal viruses: The families, subfamilies and genera of eukaryote-infecting viruses, including double-stranded (ds) DNA, single-stranded (ss) DNA, dsRNA and ssRNA reverse transcribing (ssRNA-RT) viruses, as well as prokaryote-infecting dsDNA and ssDNA viruses that have been described in literature are listed.

**Table 1 viruses-12-00832-t001:** Described viruses found in the female genital tract.

Eukaryote-Infecting DNA Viruses					
dsDNA					
*References*	*Family*	Subfamily	Genus	Species	Type
[9,12,13,14,15,16]	*Herpesviridae*	*Alphaherpesvirinae*	*Simplexvirus*	HSV-2	
			*Varicellovirus*	Varicella-zoster virus (HHV-3)	
		*Betaherpesvirinae*	*Cytomegalovirus*		
			*Roseolovirus*	HHV-6, HHV-7	
			*Proboscivirus*		
		*Gammaherpesvirinae*	*Lymphocryptovirus*	Epstein–Barr virus (HHV-4)	
			*Rhadinovirus*		
[14]	*Alloherpesviridae*		*Cyprinivirus*		
[9,10,11,12,13,15,16,17]	*Papillomaviridae* *	*Firstpapillomavirinae*	*Alphapapillomavirus*	Alphapapillomavirus 1	e.g., HPV32, 42
				Alphapapillomavirus 2	e.g., HPV77
				Alphapapillomavirus 3	e.g., HPV61, 62, 72, 81, 83, 84, 86, 87, 89, 102, 114
				Alphapapillomavirus 5	e.g., HPV**51**, *26, 82*, 69
				Alphapapillomavirus 6	e.g., HPV**56, 66**, *53*, 30
				Alphapapillomavirus 7	e.g., HPV**18, 39, 45, 59, 68**, *70*, 97
				Alphapapillomavirus 8	e.g., HPV7, 40, 43, 91,
				Alphapapillomavirus 9	e.g., HPV**16, 31, 33, 35, 52, 58**, *67,*
				Alphapapillomavirus 10	e.g., HPV6, 11, 13, 44, 74
				Alphapapillomavirus 11	e.g., HPV*73*, 34
				Alphapapillomavirus 13	e.g., HPV54
				Alphapapillomavirus 14	e.g., HPV7, 90, 106
			*Betapapillomavirus*		
			*Gammapapillomavirus*		
			*Deltapapillomavirus*		
			*Dyoetapapillomavirus*		
			*Dyothetapapillomavirus*		
			*Phipapillomavirus*		
		Unclassified *Papillomaviridea*			e.g., HPV-85
[9,12,15,17]	*Polyomaviridae*		*Alphapolyomavirus*	Human polyomavirus-5 (MCPyV)	
			*Betapolyomavirus*	Human polyomavirus-1 (BKPyV), -2 (JCPyV),	
[9,12]	*Adenoviridae*		*Mastadenovirus*	Human adenovirus B and D	
[12,14]	*Poxviridae*		*Molluscipoxvirus*	Molluscum contagiosum virus-1 and -2	
[14]	*Phycodnaviridae*		*Chlorovirus*		
[14]	*Mimiviridae*		Unclassified *Mimivirus*		
[14]	*Iridoviridae*		*Betairidoviridae*	Iridovirus	
[14]	*Marseilleviridae*			Marseillevirus	
**ssDNA**					
[9,12,15,16,17]	*Anelloviridae*		*Alphatorquevirus*		
			Unclassified *Anellovirus*	Torque teno virus, SEN virus	
[16]	*Genomoviridae*		*Gemykibivirus*		
**Eukaryote-Infecting RNA Viruses**					
**dsRNA**					
[13]	*Partitiviridae*				
**ssRNA-RT**					
[18,19,20]	*Retroviridae*	*Orthoretrovirinae*	*Lentivirus*	HIV-1	
**Prokaryote-Infecting DNA Viruses**					
**dsDNA**					
[14]	*Podoviridae*				
[14]	*Siphoviridae*				
[14]	*Myoviridae*	*Tevenvirinae*	*Tequatrovirus*	T4 virus	
**ssDNA**					
[14]	*Microviridae*				

* Including only subfamilies, genera, species and types that have been described to infect the genital mucosa. Risk of HPV types is indicated using different font types: **high-risk**, *possibly high-risk* and low risk.

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
