# Peer review of "The Vaginal Virome—Balancing Female Genital Tract Bacteriome, Mucosal Immunity, and Sexual and Reproductive Health Outcomes?"

_viruses, 2020, doi:10.3390/v12080832_

Round 1

Reviewer 1 Report

The article is confusing. There is no description of the methodology that was used. I did not understand the purpose of the study. The data found are not well discussed and need to be evaluated with other findings in the literature. It is a very poor article because it is a review article.

Summary: Not concise with the rest of the text.

Introduction: It has no clear and defined objective

What methodology was used?
How did the authors review the selected articles?
All of these aspects must appear in the article and be well described.

Author Response

Dear Reviewer 1,

Please find below a point-by-point response to your comments raised:

1. The article is confusing. There is no description of the methodology that was used. I did not understand the purpose of the study. The data found are not well discussed and need to be evaluated with other findings in the literature. It is a very poor article because it is a review article.

Thank you for this feedback. We would like to clarify that this is a non-systematic review. We have made this clearer early on in the text (lines 77-83). The primary  goal of this review was to summarize the current knowledge of the pro- and eukaryote infecting viruses making up the vaginal virome. Secondary outcomes included describing interactions between the vaginal virome and other constituents of the vaginal microbiota, and possible associated adverse health outcomes. We have now included the objective and methodology used in the article.

2. Summary: Not concise with the rest of the text.

We have updated the abstract in order to make it more concise and reflect the discussed literature better (lines 22-36).

3. Introduction: It has no clear and defined objective
What methodology was used?
How did the authors review the selected articles?
All of these aspects must appear in the article and be well described.

We have added the primary and secondary objectives of this non-systematic review article to the introduction, alongside the methodology used (lines 77-83).

I hope that we have clarified the purpose of this review.

Kind regards,

Anna-Ursula Happel

Reviewer 2 Report

Dear Authors

Well structured review with content well defined in the different paragraphs with clear concepts.

Small additions are required in the points I point out:

Line 177-180:

In turn, HSV-1 has been shown to protect C. albicans by down-regulating monocyte-mediated anti-Candida immune system responses [50],suggesting bilateral interactions between fungal and viral communities. HIV-1 envelope and transactivating proteins have also been shown to bind to C. albicans, which may promote fungal virulence by inducing hyphae formation [51–53]. 

deepen by inserting phrases on the problem of candidiasis and resistance found as problematic to common antifungals and inserting a hint to molecules or remedies of natural origin in contrasting them, also insert the following updated bibliographic entries:

Pubmed code:

PMID: 29017356

PMID: 32344551

PMID: 30676066

Insert virulence mechanisms between lines 163 - 183 and rare of candide efflux pumps and viral co-infections, insert the following biblio entry:

PMID: 32036352

deepen the discourse of gender and age in the incidence of infertility with the following biblio items:

PMID: 12667236

PMID: 27704144

Author Response

Dear Reviewer 2,

Please find below a point-by-point response to your comments raised:

1. Line 177-180: In turn, HSV-1 has been shown to protect C. albicans by down-regulating monocyte-mediated anti-Candida immune system responses [50],suggesting bilateral interactions between fungal and viral communities. HIV-1 envelope and transactivating proteins have also been shown to bind to C. albicans, which may promote fungal virulence by inducing hyphae formation [51–53]. 

deepen by inserting phrases on the problem of candidiasis and resistance found as problematic to common antifungals and inserting a hint to molecules or remedies of natural origin in contrasting them, also insert the following updated bibliographic entries: Pubmed code:

PMID: 29017356
PMID: 32344551
PMID: 30676066

Thank you for this constructive feedback. We have updated this section in the article (from line 226)  and included the suggested additional literature, where appropriate.

2. Insert virulence mechanisms between lines 163 - 183 and rare of candide efflux pumps and viral co-infections, insert the following biblio entry:

PMID: 32036352

deepen the discourse of gender and age in the incidence of infertility with the following biblio items:

PMID: 12667236

PMID: 27704144

We have also addressed this comment (lines 299-306) and included the suggested literature.

Many thanks and kind regards,

Anna-Ursula Happel

Reviewer 3 Report

Dear Authors,

The present study "The vaginal virome – balancing female genital tract bacteriome, mucosal immunity and sexual and reproductive health outcomes?" is an interesting review manuscript.

This review showed significant information about the vaginal virome and bacteriome and also highlights the most important sexual and reproductive health outcomes. The text is properly organized and includes very interesting and important information for readership.

I have some suggestion in order to improve the manuscript, which are the following:

  • The chapter 2 should be indented from the previous text.
  • L137 please number the subchapter
  • It is advisable to introduce the references for Table 1
  • L302 please number and rename the chapter as "Conclusions and future perspectives"
  • L311 remove “6.Patents”
  • Please insert the abbreviations list 

Author Response

Dear Reviewer 3,

Thank you for your constructive feedback. Please find a point-by-point response to your comments below:

1. The chapter 2 should be indented from the previous text.

Thank you, we have corrected this (line 85).

2. L137 please number the subchapter

We have now numbered this subchapter (now in line 192, subchapter 3).

3. It is advisable to introduce the references for Table 1

We have added references for Table 1 as suggested.

4. L302 please number and rename the chapter as "Conclusions and future perspectives"

This has been done as suggested (line 415).

5. L311 remove “6.Patents”

We have corrected this error (line 425).

6. Please insert the abbreviations list 

We have included the abbreviation list in the revised manuscript (starting from line 436).

Kind regards,

Anna-Ursula Happel

Reviewer 4 Report

The authors of the paper "The vaginal virome – balancing female genital tract  bacteriome, mucosal immunity and sexual and reproductive health outcomes?" reviewed current knowledge of vaginal viroma to include both human viruses and bacteriophage.

The review is well written and accurate although it is not mentioned the role that some viral infections have in conditioning the adaptive immune response.

Author Response

Dear Reviewer 4,

Thank you for your constructive feedback. Please find below our response to your point raised:

1. The review is well written and accurate although it is not mentioned the role that some viral infections have in conditioning the adaptive immune response.

Thank you for this feedback. We have now included a paragraph on the role of some viruses for conditioning the adaptive immune response (in section 3.1, lines 245-256 and lines 267-275).

Kind regards,

Anna-Ursula Happel

Round 2

Reviewer 1 Report

The authors made the alterations and submitted a new version.